# Targeting P21-Activated Kinase-1 for Metastatic Prostate Cancer

**DOI:** 10.3390/cancers15082236

**Published:** 2023-04-11

**Authors:** Payaningal R. Somanath, Jonathan Chernoff, Brian S. Cummings, Sandip M. Prasad, Harvey D. Homan

**Affiliations:** 1Department of Clinical & Administrative Pharmacy, College of Pharmacy, University of Georgia, Augusta, GA 30912, USA; 2MetasTx LLC, Basking Ridge, NJ 07920, USA; 3Fox Chase Cancer Center, Philadelphia, PA 19111, USA; 4Department of Pharmaceutical Sciences, Eugene Applebaum College of Pharmacy and Health Sciences, Wayne State University, Detroit, MI 48201, USA; 5Morristown Medical Center, Atlantic Health System, Morristown, NJ 07960, USA

**Keywords:** prostate cancer, PAK1, SSL-IPA-3, metastasis, therapy

## Abstract

**Simple Summary:**

Metastatic prostate cancer (mPCa) results in high mortality and there are no effective treatments. The molecular mechanisms mediating the transition of early-stage PCa cells into malignant, metastatic cells are not clearly understood. Whereas the TGFβ pathway has been demonstrated to induce the epithelial-to-mesenchymal transition of prostate cancer cells, it necessitates the activation of the Rac-p21-activated kinase (PAK) signaling in this process. Identifying the role of PAK in mPCa, and compounds targeting its activity may lead to novel therapeutic applications for preventing and treating mPCa.

**Abstract:**

Metastatic prostate cancer (mPCa) has limited therapeutic options and a high mortality rate. The p21-activated kinase (PAK) family of proteins is important in cell survival, proliferation, and motility in physiology, and pathologies such as infectious, inflammatory, vascular, and neurological diseases as well as cancers. Group-I PAKs (PAK1, PAK2, and PAK3) are involved in the regulation of actin dynamics and thus are integral for cell morphology, adhesion to the extracellular matrix, and cell motility. They also play prominent roles in cell survival and proliferation. These properties make group-I PAKs a potentially important target for cancer therapy. In contrast to normal prostate and prostatic epithelial cells, group-I PAKs are highly expressed in mPCA and PCa tissue. Importantly, the expression of group-I PAKs is proportional to the Gleason score of the patients. While several compounds have been identified that target group-I PAKs and these are active in cells and mice, and while some inhibitors have entered human trials, as of yet, none have been FDA-approved. Probable reasons for this lack of translation include issues related to selectivity, specificity, stability, and efficacy resulting in side effects and/or lack of efficacy. In the current review, we describe the pathophysiology and current treatment guidelines of PCa, present group-I PAKs as a potential druggable target to treat mPCa patients, and discuss the various ATP-competitive and allosteric inhibitors of PAKs. We also discuss the development and testing of a nanotechnology-based therapeutic formulation of group-I PAK inhibitors and its significant potential advantages as a novel, selective, stable, and efficacious mPCa therapeutic over other PCa therapeutics in the pipeline.

## 1. Introduction

Prostate cancer (PCa) is a significant health concern in the developed world [1]. According to the American Cancer Society, PCa is the third-most diagnosed cancer in the U.S. in 2023, just after skin and breast cancer, and accounted for an estimated 13.1% of all new cancer cases in 2022 [2]. More than 1.4 million new cases and 375,000 deaths from PCa were reported for 2020 [2,3,4,5], making it responsible for the second most cancer deaths among men worldwide [6]. In 2023, 288,300 men in the U.S. will be newly diagnosed with PCa, and 34,700 will die of PCa [2]. One in every eight men in the U.S. will be diagnosed with PCa over their lifetimes [1]. There are an estimated 3.2 million men in the U.S. today living with PCa. One man in 41 will die of PCa, and African American men are at an increased risk for developing PCa over white men in the U.S., with men of color being 1.7 times more likely to be diagnosed and 2.7 times more likely to die from PCa [7]. PCa also creates a significant economic burden that is estimated to total over $13 billion in the U.S. annually [7].

The expected five-year survival rate for PCa is close to 100% for individuals in Stages 1, 2, and 3, where the tumor is restricted to the prostate [8]. However, the expected five-year survival rate for men in Stage 4 is less than 30%, with a median survival of 42 months. Despite several controversies, prostate-specific antigen (PSA) testing is still an early screening method for PCa and is often combined with transrectal ultrasound-guided (TRUS) prostate tissue biopsies for more accurate diagnosis and staging [9]. Approximately 7% of men have metastatic disease at the time of initial diagnosis. This poor prognosis is due in part to the high tendency of metastatic PCa (mPCa) patients to develop bone metastases, which are found in 62% of mPCa patients [10]. While research into new therapies for mPCa is growing, and some treatments slow the progression of the disease, a majority of mPCa patients have no effective treatment options [11].

p21-activated kinases (PAKs) regulate a variety of normal cellular functions, including cell survival, cytoskeletal remodeling, motility, and proliferation, all of which also occur in an uncontrolled manner in advanced cancers [12,13]. Although a promising target to prevent and/or treat metastatic cancers, pharmacologically targeting PAK has been a challenge due to the lack of specific and efficacious compound inhibitors with fewer off-targets and reduced toxicity. The current review describes PCa pathophysiology, challenges in the management of mPCa patients, current and future therapies for mPCa patients to reduce overall mortality, and discusses the potential therapeutic benefits and novel strategies for targeting PAKs in the management of advanced, mPCa patients.

## 2. PCa Pathophysiology and Current Treatments

PCa begins with one or more mutations in the prostate glandular cells, which are believed to more frequently occur in the peripheral basal cells [14]. Prostatic adenocarcinoma that develops from the glandular region slowly invades the surrounding prostate tissue and can remain for decades in the form of a nodule, and eventually spread to extracapsular tissues, lymph nodes, and bone [15]. Histological grading of PCa was initially performed using a Gleason scoring system, a technique based on the microscopic appearance, architecture, and glandular pattern [16,17]. Since 2014, the International Society of Urological Pathology (ISUP) has recommended the use of a 5-tier grading system which has been subsequently revised to incorporate the added predictive and prognostic value for percentage Gleason pattern 4, intraductal carcinoma, invasive cribriform carcinoma, and minor high-grade patterns [18]. Scientists are also developing nomograms based on the new stratified ISUP guidelines to predict individual risk of high and low-grade PCa by combining the assay of total PSA and percentage of free/total PSA in patents with prebiopsy tPSA between 2 and 10 µg/L, which will aid in the decision-making process, in particular in the case of critical comorbidities and when the digital rectal examinations are inconclusive [19]. 

Guidelines for PCa treatment from the National Comprehensive Cancer Network (NCCN), a consortium of 31 major National Cancer Institute-designated comprehensive cancer centers in the US, are well-respected and generally followed in the U.S. Current therapeutic options for Stage 1 PCa that have not spread to other parts of the body include active surveillance; surgery (usually radical prostatectomy); radiation therapy (i.e., external beam radiation therapy/EBRT, or brachytherapy); hormone therapy (also called androgen deprivation therapy/ADT, i.e., luteinizing hormone-releasing hormone [LHRH] agonists, LHRH antagonists, and anti-androgens) [9]. ADT therapy is also referred to as chemical castration when testosterone levels are reduced below 50 mg/dl. Side effects of ADT include sexual and reproductive issues, osteoporosis, bone fractures, weight gain, loss of muscle mass, diabetes, heart disease, hot flashes, mood changes, fatigue, and breast tenderness and growth. ADT drugs also commonly become ineffective due to the development of resistance [20].

NCCN Guidelines for advanced PCa without metastasis (TNM = M0) are for ADT. For M1, the recommendations are for ADT alone or with one of the following: Apalutamide, Abiraterone, Docetaxel, Enzalutamide, fine-particle abiraterone, or External Beam Radiation Therapy (EBRT) to the primary tumor for low-volume metastases. M1 castration-resistant PCa is defined as a doubling of prostate-specific antigen (PSA) levels within 10 months of physical (orchiectomy) or chemical (ADT) castration. Options after enzalutamide include Sipuleucel-T, Olaparib, Pembrolizumab, Radium-223 (for bone metastases), and Rucaparib. A more detailed account of the currently available treatments for mPCa has been recently reviewed elsewhere [21]. 

Notably, despite the FDA approval of abiraterone acetate, enzalutamide, and Radium-223 for front-line use in men with metastatic castration-resistant PCa (mCRPC) in the past five years, the life expectancy for patients with PCa over the past decade has only been prolonged by one year [22,23]. Thus, due to a lack of effective treatment options, many patients with mPCa succumb to the disease. In addition, current nomograms in high-risk and advanced prostate cancer are limited, especially in evaluating the outcomes of patients with castration resistance [24].

## 3. P21-Activated Kinases (PAKs)

PAK was discovered about 30 years ago as a specific downstream effector of a group of 21 KDa proteins, Rho GTPases (p21) in rat brain cytosol [25]. A plethora of research findings from several laboratories during the last three decades identified a 6-member family of PAKs categorized into two classes, group-I and group-II PAKs based on their mechanism of activation [12]. Group-I PAK isoforms, namely PAK1, PAK2, and PAK3, are activated by GTP-bound Rho family GTPases, Rac, and Cdc42 [26,27,28]. Group-I PAKs have been shown to regulate transcription, protein synthesis, and cellular signaling pathways in numerous cells types through serine-threonine phosphorylation of various proteins that mediate cytoskeletal remodeling, focal adhesion assembly, cell motility [29], survival [30], and cell cycle [31,32,33]. Group-I PAKs are activated by Rac and Cdc42 in cells, which in turn promotes the formation of lamellipodia and filopodia [29,34], two cellular structures critical to cell migration. Apart from Rac and Cdc42, several kinases, phosphatases, lipids, and adaptor proteins have also been implicated in PAK activity regulation [12,13].

The most prominent difference between the two classes of [35], is their mechanism of activation. Specifically, the activation of Group-II PAKs (PAK4, PAK5, and PAK6) is not dependent on Rho GTPases [35,36]. Despite these differences, there is a high similarity between the kinase domains of these six kinases [37,38]. This similarity serves as a barrier to the development of specific inhibitors by directly targeting their kinase domain. Fortunately, other structural and biochemical differences, intracellular and tissue distribution, unique substrate specificities [37], and distinct cellular functions [39,40] exist between the two groups of PAKs that can be exploited to develop selective inhibitors. 

Compared to Group-II PAKs, Group-I PAKs are more ubiquitously expressed in tissues [41]. Group-I PAKs have been identified in the brain [42], heart [43], and neutrophils [44], of which, PAK1 is the predominantly expressed isoform. In contrast, a high level of PAK3 mRNA expression has been reported in the brain [42], the pituitary [45], and many other organs. PAK2 is expressed in abundance in the heart and vascular tissue, and inhibition of PAK2 is associated with cardiac toxicity [46].

PAK4 is the best-characterized isoform among the Group-II PAKs and is abundantly expressed in the prostate, testis, lung, heart, brain, and liver [47,48]. Whereas PAK4 gene knockout in mice is embryonically lethal [49], PAK5 knockouts [50] and PAK5/PAK6 double knockout mice [51] are viable and lack any obvious phenotype, suggesting the critical importance of PAK4 among the Group-II PAKs. Although the brain expresses all PAK isoforms, including PAK5 [52], PAK6 expression is restricted to the testis, prostate [53], brain, kidney, and placenta [37,54]. Although PAK1 is the most ubiquitously expressed isoform among all six PAKs, it is undetected in the normal prostate gland [55], suggesting that the cellular expression and tissue distribution of PAKs are highly restricted.

## 4. Structure and Mechanism of Activation of Group-I PAKs

### 4.1. Activation of PAKs by the GTPases

The structure and function of Group-I PAKs have been evolutionarily conserved [56,57] and these proteins share close similarities in their N-terminal regulatory domain and a C-terminal catalytic domain [58,59,60]. In human PAK1, GTP-bound Rac and Cdc42 family members interact with a conserved N-terminal non-catalytic domain known as the p21 binding domain (PBD) ranging from amino acids 67 to 113 [58,59,60] (Figure 1). The Group-I PAK regulatory region also consists of an auto-inhibitory domain (AID) (amino acids 83 to 149) that partially overlaps with the PBD region [59]. Several adaptor proteins such as Nck and Grb2, and the guanine nucleotide exchange factors (GEFs) PAK-interacting exchangers/cloned out of a library (PIX/COOL) bind to a highly conserved proline-rich region in the regulatory domain of Group-I PAKs [61,62,63]. The catalytic domains of human group-I PAKs are highly conserved and consist of an acidic region rich in Glu/E and Asp/D in between the N-terminal domain and catalytic domain [35,64]. In contrast, Group-II PAKs employ a different mechanism of autoregulation that involves a pseudosubstrate interaction with the catalytic domain.

Years of research by various research groups on the structural [59], genetic [64], and biochemical [65,66] characterizations of PAK isoforms have yielded essential insights into the involvement of GTPases in the activation of group-I PAKs. Among the GTPases, Rac isoforms such as Rac1, Rac2, and Rac3 [25,58], Cdc42 [25] as well as CHP/Wrch2/RhoV [67], TC10/RhoQ [68], and Wrch1/RhoU [69] have been identified as activators of group-I PAKs (Figure 1). The fact that GTP-bound Rac and Cdc42 are essential for PAK1 binding and activation was confirmed by the increased effector affinity of Rac and Cdc42 with activating mutations (Q61L) that convert them into GTP-hydrolysis deficient variants [29,34,70,71].

Although the interaction of αPIX, a member of the PIX/COOL family of GEFs, with PAK1, has been demonstrated to induce PAK1 activation by Rac or Cdc42 [72], the mechanism by which αPIX activates PAK1 is unknown. βPIX, another PIX family member, is reported to modulate PAK1 activity [62,72], but reports on whether the interaction between βPIX and PAK enhances or inhibits its activity is conflicting [73,74] (Figure 1).

PAK homodimers are inactive in the resting stage, due to a trans-inhibitory switch involving the interaction of the AID with the kinase domain of the dimer partner. The binding of GTP-bound Cdc42 or Rac1 with the interactive binding (CRIB) region of PAK1 (amino acids 75 to 90) changes the protein conformation disrupting their dimeric interaction and removing the trans-inhibitory switch [70,75,76]. The specificity of the GTPase interaction with the CRIB domain is provided by a short upstream lysine-rich region in human PAK1 (amino acids 66 to 68) [58,60]. The fact that GTP-bound Rac and Cdc42 are essential for PAK1 binding and activation was confirmed by the increased effector affinity of Rac and Cdc42 with activating mutations (Q61L), which converted PAK1 into GTP-hydrolysis deficient variants [29,34,70,71]. The interaction of GTP-bound Rac and cdc42 with the adjacent PBD region further strengthens this binding and activation of PAK [59], which is preceded by the release of the AID and autophosphorylation at Thr423 in the catalytic domain [77]. Studies suggest that the interaction between the PAK-PBD domain and GTP-bound GTPase may likely act as an allosteric mechanism of PAK1 activation [59,66]. Whereas one study in vivo reported a potential role for 3-phospho-inositide dependent kinase-1 (PDK1) in Thr423 phosphorylation [77], others suggest that autophosphorylation of PAK1 at Ser144 and PAK2 at Ser139 also plays a role in maximizing PAK activity [25,65,66].

Intra-cellular translocation of PAKs from one compartment occurs upon various extracellular stimuli [78]. Although some studies show the interaction of GTPases with Group-II PAKs, this, however, does not result in their activation but likely is necessary for their intracellular transport [79]. Interestingly, a mutant of Cdc42 (Y40C), which cannot interact with the group-I PAKs, retains its ability to interact with group-II PAKs [47]. Once group-I PAKs are activated, the GTPase binding to their CRIB and PBD regions is not essential to prolong their kinase activity [25]. 

### 4.2. GTPase-Independent Activation of Group-I PAKs

Whereas the Rho family GTPases have been considered the major mediators of group-I PAK activation, mechanisms independent of GTPases have also been implicated in PAK activation. The proline-rich N-terminal (PXXP) region of PAK binds SH-3-containing adapter proteins such as Nck and Grb2 [80,81]. This binding is necessary for the recruitment of group-I PAKs to the plasma membrane and activation by PDK1 that phosphorylates the conserved Thr423 of the PAK1 activation loop [32,82]. 

Sphingolipids and their derivatives have been shown to mediate PAK activation by facilitating their recruitment to the membrane [82]. Membrane recruitment of PAK has also been linked to receptor-tyrosine-kinase-mediated responses [80,81]. The Akt family of kinases that enhances PAK1 activity has been proposed to phosphorylate Ser21 of PAK1 [83], which in turn, reduces the affinity of Nck binding with PAK1 thus affecting cell motility [83]. Etk/Bmx, a non-receptor tyrosine kinase, has been implicated to directly phosphorylate PAK1 [84], although its role in PAK1 activation is not clear. The importance of PAK1 phosphorylation at a tyrosine residue has been reported to mediate the synergistic activation of breast cancer cells by ERα and estrogen [85]. 

Apart from several kinases, PAK activity has also been reported to be regulated by serine-threonine phosphatase PP2A by directly interacting with PAK [86,87]. In addition, Pix1 and Pix2 also have been demonstrated to inactivate PAK [87] by interacting with the Pix family of PAK-specific GEFs, forming a multi-protein complex involving PAK, and dephosphorylating Thr423 of PAK1 [87]. More recently, RIT1, a small GTPase similar to Ras has been demonstrated to control actin dynamics in COS7 cells by directly binding to RAC/Cdc42 bound PAK1 [88].

## 5. PAK1-Mediated Regulation of Cytoskeletal Remodeling and Cell Motility

Small GTPases such as Rho, Rac, and Cdc42 are major regulators of cellular cytoskeletal dynamics [32]. Among these proteins, GTP-bound Rac and Cdc42 achieve their effects partly through PAKs, which phosphorylate and interact with various proteins regulating cytoskeletal remodeling [35,36,86]. Substrates of PAKs include, but are not limited to, several adaptor proteins, guanine exchange nucleotide factors, intermediate filaments, microtubules, integrins, kinases, and phosphatases [32,89]. 

PAKs have also been demonstrated to modulate myosins, a large family of actin-based molecular motor proteins, which alters cell spreading, migration, and proliferation [90]. Ste20 and Cla4 in *Saccharomyces cerevisiae*, two PAK-related kinases that are highly homologous to the group-I PAKs, are among the first identified to phosphorylate myosin light chain (MLC) by Rac and Cdc42 activation [91,92]. PAKs have also been implicated in the direct activation of myosin light chain kinase (MLCK) by phosphorylation [93]. 

In mammals, PAK phosphorylation of neuronal MLC on Ser19 results in actin stabilization through a GIT1/PIX/Rac/PAK complex formation [94]. PAK1 regulates branching morphogenesis in a 3D culture of MDCK cells by a PIX/ integrin β1-dependent mechanism [95]. Fibrinogen-induced PAK1/Cofilin Pathway activation has also been reported in endothelial cells [96]. The Dishevelled, EGL-10 and Pleckstrin (DEP) domain-containing protein 1B has been reported to promote pancreatic cancer cell motility and invasion via activation of the PAK1/LIMK1/Cofilin1 Signaling Pathway [97]. The MICAL1 monooxygenase, an important regulator of filamentous actin (F-actin) is activated by PAK1 to mediate actin disassembly in HEK293T cells [98].

LIM Kinase (LIMK), a serine-threonine kinase involved in the regulation of actin cytoskeletal assembly and microtubule disassembly is another known substrate of PAKs. LIMK phosphorylates PAKs at their Thr508 residue [99]. PAK-activated LIMK phosphorylates cofilin, an actin-capping and severing protein [100], which promotes lamellipodia and filopodia formation, and cell migration [32]. Filamin A, an actin-binding protein regulating the cross-linking of the actin network, interacts with the CRIB region of PAK1 resulting in filamin A phosphorylation (Ser2152) [101]. In addition, PAK is also involved in the phosphorylation of p41-Arc (Ser21), a 41 kDa subunit of actin-related protein 2/3 (Arp2/3) complex [102], cortactin, an F-actin-binding protein (Ser113) involved in actin polymerization [103] and caldesmon, an actin filament regulatory protein (Ser657 and Ser687) [104] that promotes actin polymerization. An association of PAK1 with the actin binding protein filamin A enables vimentin phosphorylation and filament assembly, which are important in the development and stabilization of cell extensions during cell migration [105].

PAK1 has been suggested to phosphorylate integrin-linked kinase (ILK) (Thr173 and Ser246) mediating its nuclear export [106], the functional significance of which is still unclear. Other PAK-regulated proteins involved in the actin cytoskeletal remodeling and cell motility are armadillo adaptor protein (Ser561 and Ser688), a Drosophila counterpart of mammalian β-catenin [107], paxillin, an adaptor protein involved in integrin-dependent fibrillar adhesions [108]. Phosphorylation of paxillin S273 is a key event in the regulation of cell migration through the recruitment of βPIX and PAK1 to sites of adhesion [109].

PAKs also phosphorylate several proteins involved in microtubule dynamics. Stathmin (oncoprotein 18 or op18) is known to cause microtubule disassembly [110]. PAK phosphorylates op18 (Ser16) leading to microtubule stabilization specifically in the migrating ends of cells. Another mechanism by which PAK1 stabilizes tubulin is via phosphorylation of tubulin cofactor-B (TCoB at Ser65 and Ser128), which promotes the heterodimerization of α/β-tubulins [111]. PAK is also involved in the phosphorylation of dynein light chain-1 (DLC1 at Ser88), which provides strength to the microtubules [112]. GIT1/βPIX signaling proteins with PAK1 also represent a novel regulatory mechanism of microtubule nucleation in interphase cells [113]. During proplatelet extension, PAK1 regulates MEC-17 acetyltransferase activity and microtubule acetylation [114]. A recent report suggests a novel role for PAK1 in regulating cell shape through ribonucleoprotein granules during normal and stressed growth conditions [115]. In breast cancer cells, prolactin-induced PAK1 activation results in augmented cell motility [116,117].

## 6. PAK Signaling in the Endothelium and Tumor Blood Vessels

Adaptive and tumor neovascularization is reliant on endothelial cell motility [118]. This, in turn, is regulated by the Rho family of GTPases [119] involving PAK1 [120]. The importance of PAK1 and PAK2 in endothelial cell motility has been demonstrated in mouse gene knockout studies [29,121,122,123,124,125]. Interestingly, impaired actin cytoskeletal remodeling and cell motility in fibroblasts deficient in Rac1 or Akt1 were reversed by overexpression of a mutant PAK1 (T423E) with constitutive activation [29,121,122]. Rac-PAK1 signaling is important for the directional migration of endothelial cells [126], the inactivation of which results in the impaired cytoskeletal assembly, lamellipodia formation, and migration in vitro [34,127,128]. Furthermore, PAK1 is important in the regulation of endothelial cell survival, cell cycle, and proliferation [129,130]. Overexpression with a dominant-negative p65 PAK peptide that disrupts the interaction between PAK and Nck in endothelial cells prevented motility and tube formation in a 3D Matrigel model in vitro [131] and a chick allantoic membrane (CAM) angiogenesis assay in vivo [132]. PAK has also been implicated in integrin activation, focal adhesion turnover, maintenance of endothelial barrier integrity [133], and tube formation [134]. The role of PAK1 in neovascularization has also been demonstrated in a zebrafish model [135]. Although the role of PAK1 in tumor angiogenesis has not been studied in detail, one study does suggest a role for PAK1 in the regulation of tumor angiogenesis in ovarian cancer [136]. A recent study in glioblastoma reports augmented angiogenesis involving VEGF-mediated activation of PAKs and ERK signaling pathways [137]. Our study in a PCa tumor xenograft model demonstrated impaired tumor angiogenesis with PAK1 or PAK6 gene knockdown, with PAK1 suppression exhibiting a superior anti-angiogenic effect than PAK6 suppression [55]. This study, however, was an effect of PAK1 and PAK6 gene knockout in PCa cells on tumor angiogenesis, and not an effect of direct suppression of PAKs in endothelial cells. A previous finding in breast cancer cells suggests that decreased tumor angiogenesis in PAK1-deficient PCa tumor xenografts could be caused by reduced VEGF production by the PAK1-deficient PCa cells [138].

Vascular permeability-inducing effects of thrombin involve activation of Rac/PAK signaling, which in turn, disrupts the endothelial barrier junctions [139]. Similarly, VEGF-induced endothelial monolayer permeability is mediated by Rac-induced internalization of VE-cadherin [140]. Furthermore, endothelial cell activation by TNFα, bFGF, and histamine resulted in Rac activation, PAK1 phosphorylation at Ser141, and the translocation of PAK1 to endothelial cell barrier junctions [141]. In this process, PAK1 has also been implicated in directly phosphorylating VE-cadherin to destabilize the endothelial cell junctions [140]. A recent study has identified PAK1 to contribute to tumor angiogenesis through the HIF1α/VEGF pathway [142]. Collectively, these studies suggest potential anti-angiogenic and anti-vascular permeability effects of PAK1 inhibition. 

Cancer cells need to degrade and remodel the extracellular matrix (ECM) to escape from their primary niche [143]. Intriguingly, matrix metalloproteinases are found to be abundantly expressed in advanced tumors, which are often degraded by the cancer cell-secreted proteases [144]. Remarkably, increased pericellular proteolysis observed in a model of pre-malignant progression of breast cancer was reliant on increased PAK1 activity [145]. This suggests that PAK1 suppression may improve tumor resistance by the stroma in addition to disabling the cancer cell proliferation, vascular invasion, and metastasis.

## 7. PAK1 in Cancer

Cancer cells must proliferate and survive in an environment that lacks their pro-growth signals for solid tumors to increase in size and metastasize. Additionally, cancer cells from these tumors must resist inhibitory growth signals present in the surrounding stroma and still find a way to link to blood vessels for nutrients and oxygen. Once attached to blood vessels, cancer cells must find a way to enter the blood supply, and then escape these same vessels at a distant site. Cancer cells need to do all of this to metastasize while maintaining a higher rate of metabolism and protein synthesis. For solid tumors to become established, grow, and spread, the cancer cell must proliferate, overcome the inhibitory effects of surrounding stroma and survive in an environment that lacks its normal survival factors. Hence, tumors must co-opt and attract the blood vessels for the continuous supply of nutrients and oxygen, escape the primary site of origin, and maintain a higher rate of metabolism and protein synthesis [12,146]. 

Group-I PAKs are reported to be involved, either directly, or indirectly, in the modulation of the various steps in tumor initiation and progression describe above. As such, PAKs make an attractive therapeutic target for cancer. A comprehensive analysis of the prognostic implications and functional exploration of the PAK gene family in human cancer from the Cancer Genome Atlas (TCGA) and Genotype-Tissue Expression (GTEx) databases supports the prognostic and therapeutic potential of PAKs [147]. Increased PAK1 activity and expression have been linked to several human malignancies, primarily cancer [148]. PAK1 abnormalities such as gene amplification leading to increased mRNA and protein expression, phosphorylation, and increased accumulation of the activated form of this enzyme [149] have been linked to tumor prognosis [150]. The accumulation of active PAK1 in the nuclei of malignant cells [151] and a causal link to breast cancer progression in mice have been demonstrated [152,153]. PAK1 has also been implicated in promoting metastasis in triple-negative breast cancer [154]. Pharmacological inhibition of PAK1 suppresses breast cancer motility and invasion [155] and potentiates the effects of microtubule-stabilizing agents [156], restoring tamoxifen sensitivity in breast cancer cells [157]. A role for Rac1/PAK1/miRNA142 signaling in tumorigenesis has further been reported in breast cancer cells [158] (Table 1). 

Whereas overexpression of PAK1 was identified as an independent prognostic marker of poor outcomes in ovarian cancer [151], it is a nuclear expression and phosphorylation (Ser305) in breast cancer cells that predict resistance to tamoxifen therapy. The cytoplasmic levels of PAK1 correlated with recurrence rate and mortality in breast cancer [159,160]. Overexpression of the constitutively active form (T423E mutant) of PAK1 in breast cancer cell lines stimulated anchorage-independent growth in vitro [161], resulting in mammary epithelial hyperplasia [162] and, breast carcinomas [152] in mice (Table 1). 

PAK1 has been associated with gastric cancer cell migration, invasion [163], and hematogenous metastasis [164]. In gastric cancer patients, high PAK1 expression was associated with advanced-stage tumors, distant metastasis, and reduced survival [165,166]. Recent studies also indicate PAK1 activation by several long non-coding RNAs and miRNAs in the progression and EMT of gastric cancers [167,168]. In a mouse model of intestinal cancer, PAK1 has been demonstrated to initiate tumor development [169]. Another recent study associated PAK1 with the immune response to intestinal tumors [170]. PAK1 activation was linked to autophagy and immune evasion in pancreatic ductal carcinoma [171]. 

Exosome-derived miR-485-3p from normal pancreatic ductal epithelial cells inhibited pancreatic cancer metastasis by directly targeting PAK1 [172]. The benefits of co-targeting PAK1 signaling with other chemotherapeutic agents have also been shown to improve the therapeutic outcome of pancreatic cancers [173,174] and decrease pancreatic cancer cell resistance to treatments [175]. 

In non-small lung cancer cells, PAK1 conferred chemoresistance and poor outcomes [176]. Similarly, the activation of PAK1/AKT signaling promoted resistance to FGFR1 inhibition in squamous-cell lung cancer [177]. Deregulated PAK1 expression has correlated with aberrant epithelial-to-mesenchymal transition (EMT) marker expression and poor prognosis in non-small cell lung cancer [178]. These observations suggest a causal role for PAK-1 in non-small lung cancer.

Elevated PAK1 has been correlated to cancer cell infiltration and metastasis in colorectal cancer, [179]. This was likely due to the facilitation of EMT [148]. Other studies have indicated that PAK1 activation promotes metastasis of osteosarcoma [180], as well as Ra1-PAK1 signaling in EMT in human hepatocellular carcinoma [181] (Table 1). 

Kinesin superfamily protein 4-mediated activation of Rac1/PAK1 signaling has been reported to induce cytoskeletal changes in glioma cells [182]. Importantly, PAK3 has been reported as a signature gene of the glioma pro-neural subtype affecting proliferation, growth, and differentiation [183]. In melanoma cells, RAC1/PAK1 pathway activation promotes their metastasis to the brain [177].

In human bladder cancer, PAK1 expression was significantly higher than in human normal urothelial cells and normal bladder tissues, respectively [184], which was comparable to high histological grade, lymph node metastasis, and tumor size. Rac/PAK1 pathway activation has been associated with lymphovascular invasion and lymph node metastasis of upper urinary tract cancer [185]. Increased PAK1 activation has also been correlated with the histological grade and lymph node metastasis of bladder cancer [186,187]. 

PAK1 expression is also correlated to poor prognosis and immune evasion in metastatic renal cell carcinoma patients [188]. PAK1 dictated stem-like phenotype and sunitinib resistance in renal cell carcinoma cells with its reversal by treatment with IPA-3 [189], which determines its metastatic ability. These studies have demonstrated that targeting PAK1 can be an attractive strategy to treat urological cancers (Table 1).

**Table 1 cancers-15-02236-t001:** PAK1 expression in various cancers.

Cancer Type	PAK1 Expression	References
Breast Cancer	Accumulation of active PAK1 in the nucleus, cell motility, and cytoskeletal dynamics, and promotes cancer growth and metastasis of triple-negative breast cancer. Increased PAK1 expression in patient biopsies correlated to drug resistance and increased mortality.	[151,152,153,154,155,156,157,158]
Ovarian Cancer	Increased PAK1 expression correlated with poor outcomes in patients.	[151]
Gastric Cancer	Promotes migration, invasion, and hematogenous metastasis. High PAK1 expression is associated with EMT, advanced-stage tumors, distant metastasis, and reduced survival in patients.	[163,164,165,166,167,168,169,170]
Pancreatic Ductal Carcinoma	PAK1 activation is linked to autophagy and immune evasion. Co-targeting PAK1 sensitizes cancer cells to chemotherapeutics in vitro.	[171,172,173,174,175]
Non-Small Cell Lung Cancer	PAK1 deregulation promoted EMT and conferred chemoresistance and poor outcomes in patients.	[176,177,178]
Colorectal Cancer	PAK1 expression correlated to cancer cell infiltration and metastasis in patients.	[179]
Osteosarcoma	PAK1 activation promoted metastasis in patients.	[180]
Hepatocellular Carcinoma	PAK1 activation promoted EMT and metastasis in patients.	[181]
Glioma	Induced cytoskeletal changes in cells, PAK3 has been reported as a signature gene of the glioma cells in promoting proliferation, growth, and differentiation.	[182,183]
Melanoma	PAK1 activation promotes cell metastasis to the brain in mice	[177]
Bladder Cancer	Increased PAK1 expression in cancer cells compared to normal bladder epithelial cells, and was also comparable to high histological grade, lymph node metastasis, and tumor size in patients.	[184]
Upper Urinary Tract Cancer	Increased PAK1 activity associated with lymphovascular invasion and lymph node metastasis in patients.	[185,186,187]
Renal Cell Carcinoma	PAK1 expression correlated to poor prognosis, immune evasion, and metastasis in patients. PAK1 dictated stem-like phenotype and sunitinib resistance in cells with its reversal by treatment with IPA-3.	[188,189]
Lymphoma	Increased PAK1 expression develops drug resistance in lymphoma patients.	[190]
Acute and Chronic Myeloid Leukemia	PAK1 suppression inhibited cell proliferation in vitro.	[191,192,193]
Prostate Cancer	PAK1 is absent in normal prostate and prostatic epithelial cells but is expressed in PCa cells and correlates with increased invasive potential in cells and Gleason Score in patients. PAK1 suppression by IPA-3 and/or SSL-IPA3 inhibits tumor growth, EMT, metastasis to the lungs, and PCa cell-induced bone remodeling.	[55,194,195,196,197]

The importance of PAK1 in hematological cancers also has started to emerge. PAK1 has been reported to mediate drug resistance in lymphomas [190], and targeting PAK1 suppressed the proliferation of chronic myeloid leukemia cells [191]. Another recent study reports PAK1 as a therapeutic target in acute myeloid leukemia and myelodysplastic syndrome [192]. Group-I PAKs are also important for leukemia cell adhesion to matrix proteins such as fibronectin [193]. These studies hold the promise of targeting PAK1 for hematological cancers in addition to many solid cancers.

## 8. PAK1 in mPCa

Although ubiquitously expressed in all the other organs, group-I PAKs are undetected in normal prostate tissue [55]. Instead, the prostate expresses group-II PAKs in abundance, predominantly PAK4, and PAK6 [53,55,198,199]. Interestingly, a study conducted in our laboratory demonstrated that activation of Rac1-GTPase promoted PCa cell (PC3 and LNCaP)-ECM interactions, lamellipodia formation, cell migration, and trans-endothelial migration [71]. This finding was confirmed by others who showed suppression of the PCa cell cycle and cell survival by pharmacological inhibition of Rac1 [200]. This supported the hypothesis that even though group-I PAKs are not detected in normal prostate tissue, they still may be necessary for prostate tumor progression and metastasis. This hypothesis is further supported by the fact that group-I PAKs are absent in normal human prostatic epithelial (RWPE-1) cells, but are detected in human PCa cell lines [55]. Whereas in agreement with their correlation with Gleason Scores, PAK1 protein expression was the lowest in less-invasive LNCaP cells but was proportionally higher in the relatively more aggressive metastatic PCa cells such as PC3, LNCaP C4-2, and Vertebral Cancer of the Prostate (VCaP). This supports a role for PAK1 in PCa cell invasion and metastasis. This same study also showed that PAK6 is expressed in a variety of human PCa cells as well as the normal prostatic epithelial cells, and unlike PAK1, their expression is not dependent on the aggressiveness of the PCa cells [55]. 

Further support for a role for PAK1 in advanced prostate cancer can be seen by the fact that higher levels of PAK1 mRNA expression in PCa patients correlate with higher Gleason scores, as compared to low-grade PCa [195]. Additionally, increased PAK1 expression is seen in patients with PCa spread to lymph nodes and surrounding tissues [195]. Similar findings correlating PAK1 expression and PCa severity were also reported in human PCa biopsies [55]. Moreover, PAK1 expression was significantly higher in PCa metastatic lesions than in the primary tumors 55]. These findings were independently confirmed by another group that demonstrated the PAK1 upregulation in PCa associated with mTOR-mediated tumor autophagy [201]. A recent study also reported that PCa cell proliferation and EMT induced by the metastatic promoter DEPDC1B involves Rac1-PAK1 signaling [202]. Similarly, the cytotoxic necrotizing factor-1 promoted PCa cell migration and invasion by activating the Cdc42-PAK1 pathway [203]. Thus, PAK1 may be a potential tumor marker and therapeutic target of PCa. This suggestion is supported by the finding that shRNA-mediated PAK6 knockdown had no significant effect on PCa cell motility, while PAK1 knockdown significantly impaired the motility of the PC3 cells on ECM proteins such as fibronectin, vitronectin, laminin, osteonectin, and osteopontin. Furthermore, treatment with the PAK1-specific inhibitor IPA-3 significantly blunted EGF-mediated cell migration in PC3 cells [55]. 

While the above studies support the suggestion that PAK-1 expression is higher in more aggressive prostate cancer, the mechanisms involved are not fully understood. Studies have shown that activation of PAK2 in PCa cells by a proteinase inhibitor, α2-macroglobulin leads to the phosphorylation and activation of LIM domain kinase [204]. Additionally, a role for PAK1 in the regulation of myosin II-B in PCa cells in response to epidermal growth factor (EGF) was also reported revealing the direct role of PAK1 in the regulation of both light and heavy chains of non-muscle myosin II-B [205]. These studies indicated the importance of PAK1 signaling in early and advanced-stage PCa cells, possibly by mediating changes in cell structure, size, and EMT.

Increased PAK1 activity in PCa may be needed to promote EMT. This hypothesis is supported by findings from our laboratory showing that prolonged treatment of PCa cells with TGFβ1 results in further activation of the Rac1/PAK1 pathway [196]. PAK1-deficient PC3 cells were resistant to TGFβ1-induced EMT and invasion of PCa cells. IPA-3, a selective allosteric PAK1 inhibitor was able to reverse TGFβ1-associated EMT, suggesting the indispensable role of the “Rac1– PAK1” signaling axis in TGFβ1-mediated cytoskeletal remodeling and EMT in PCa cells. PAK1-deficient PC3 cells were also resistant to TGFβ1-promoted EMT and invasion of PCa cells. These results suggest that targeting this axis could be a potential treatment target for advanced-stage PCa [196] (Figure 2).

## 9. PAKs and Androgen Signaling in PCa

The prostate gland requires androgen (testosterone) for its development and optimal function [146,206], a reason why androgen-deprivation therapy (ADT) is highly effective [207]. Unfortunately, advanced-stage, castration-resistant PCa cells have been identified to produce testosterone intracellularly [208] and develop resistance to ADT via other mechanisms such as gene or enhancer amplification, receptor mutations, receptor variants, and coactivator overexpression of the androgen receptors (AR) [21]. Among the PAK isoforms, PAK6, and group-I PAK, were amongst the earliest PAKs associated with the AR [54]. This was later confirmed in both prostate and breast cancers [209,210]. PAK6 was also reported to inhibit PCa by phosphorylating the AR and tumorigenic E3 ligase murine double minute-2 (Mdm2) [211] and via direct interaction with AR [212]. No other isoforms of PAK, and none from the group-I PAKs, have ever been directly or indirectly linked to androgen signaling. Thus, inhibitors of group-I PAKs, especially PAK1, are an attractive strategy for PCa therapy without interfering with the androgen system.

## 10. Inhibitors of PAKs and Their Potential Clinical Utility

PAK1 phosphorylates and modulates the activities of a plethora of substrates ranging from cytoskeletal proteins and signaling molecules to transcription factors that regulate numerous pathways implicated in PCa cell growth [12]. Despite the availability of several PAK1 inhibitors, most of these compounds have failed in pre-clinical studies [213]. Reasons for this include their off-target effects, poor pharmacokinetic properties, and toxicity. Although a limited number of PAK inhibitors have been tested in clinical trials, almost all had to be prematurely terminated. As of today, no PAK1 inhibitor has progressed beyond stage I cancer clinical trials. These major classes of PAK1 inhibitors and their effects on cancer cells are described below.

### 10.1. ATP-Competing PAK1 Inhibitors

The first characterized PAK inhibitor was K-252a, an indolocarbazole alkaloid with a Ki value of 2.4 nM but very weak anti-proliferative activity in vitro [214]. Although derivatives of K-252a such as KTD606 and CEP1347 were developed, these compounds were only effective in an NIH-3T3 model and not in other cells [215]. Although staurosporine which consists of an indolocarbazole scaffold was effective in inhibiting PAK at 0.75 nM, its poor kinase selectivity hindered its clinical utility [216]. Next, 5-substituted monocyclic aminopyrazoles were developed by Pfizer, where the ATP adenine mimetic aminopyrimidine scaffold was expected to function as an ATP competitor [13]. Pfizer also developed PF-3758309, a bicyclic aminopyrazole as a pan-PAK inhibitor with a Ki value of 14 nM, which was moved to Phase I clinical trial for advanced-stage solid tumors (NCT00932126). However, this study was terminated due to undesirable PK characteristics, including unfavorable levels of the drug in the plasma and the lack of a dose-response association [217]. PF-3758309 was also found to inhibit PAK4, a group-II PAK isoform [218]. Genentech later developed another series of aminopyrazole-type PAK1 inhibitors, and Il-11 among them exhibited the most potent PAK1 inhibitory effect [219].

Afraxis, initially on its own and then in collaboration with Genentech, developed 2-amino pyrido [2,3-d]pyrimidine-7(8H)-one derivative as a new class of ATP-competitive PAK1 inhibitors [156]. These were found to be effective in preclinical models of NF2, KRAS-driven squamous cell carcinoma, and HER2- driven breast cancer [220,221] but the studies were terminated in 2016 due to cardiovascular toxicity [46]. FRAX597, a potent PAK inhibitor with an IC_50_ value of 7.7 nM was more selective for PAK1 over PAK4 and exhibited potent anti-proliferative effect against NF2-deficient schwannoma cells and significant antitumor activity in an orthotopic model of neurofibromatosis-2 (NF2) [220]. FRAX486 a derivative of FRAX597 was able to inhibit prostate stromal cell growth [222]. G555 is a PAK1 inhibitor from the same generation with an IC_50_ value of 3.7 nM, which was primarily used in elucidating the cellular functions of PAKs in cells [223].

The next generation of PAK1 inhibitors was based on aminopyrimidine, the most potent among them being 2-arylamino-4-aryl-pyrimidines. These showed potent inhibitory activity against PAK1, but the cellular effects are yet to be determined. To reduce the toxicity, a pyridone side chain analog G-9791 was developed as a PAK1 inhibitor but was found to be less efficacious in cells [46].

Other ATP-competitive PAK1 inhibitors include OSU-03012, a previously characterized PDK1 inhibitor derived from celecoxib, which displayed PAK1 inhibitory activity with an IC50 value of 7.7 nM [224]. AK963 is a urea derivative PAK1 inhibitor that suppressed the proliferation of gastric cancer cells by inhibiting the PAK1-NF-κB-cyclinB1 pathway [225]. Another compound, ZMF-10 presented an IC50 value of 174 nM with a good selectivity to inhibit PAK1 with a significant effect in inducing ER-Stress and suppressing cell migration by inhibiting Akt and Erk pathway and FOXO3 activation [226]. The use of these compounds was limited to the molecular characterization of PAKs in cellular responses and did not progress into clinical development likely due to their off-target effects.

Although G-5555 [223], FL172 [227], PF-3758309 [217], and AZ13705339 [228] have demonstrated PAK1 activity suppression in vitro, these compounds also exhibited off-target effects on the Src family of kinases, Akt1, AMPK, Cyclin-dependent kinase-7, and serum glucocorticoid kinase due to close similarities in their ATP-binding domains [219,229]. Therefore, allosteric PAK1 inhibitors such as IPA-3 [230] and NVS-PAK1-1 [231] may be advantageous for pharmacological interventions in cancer research (Table 2).

### 10.2. Allosteric Modulation of PAK1 Activity for PCa Therapy

A process to screen compounds that can allosterically inhibit PAK1 identified IPA-3 (Inhibitor of PAK1 Activation-3) as a small molecule allosteric PAK1 inhibitor, which was then proposed as a potential therapeutic agent for diseases with deregulated PAK1 activity, including cancer [230]. IPA-3 is unique over other PAK1 inhibitors in that it has been reported to block PAK1′s auto-phosphorylation and kinase activity through covalent interactions with an N-terminal, the non-catalytic segment of the protein, which, accounts for its high degree of specificity for PAK1 over other protein kinases [230,238]. IPA-3 does not target the ATP-binding domain of any kinase and is unique among the other inhibitors because of its allosteric nature in suppressing Group-I PAK activity. IPA-3 exhibits some selectivity for PAK1 over PAK2 and no inhibitory effects on the activity of distantly related Group-II PAK isoforms or any of the more than 200 other kinases tested in vitro [229]. Due to these properties, IPA-3 is highly selective in suppressing PAK1 activity in various cell types, hence an attractive therapeutic for cancer therapy.

However, IPA-3 has a very short half-life as it is metabolically unstable once administered in vivo [219]. A shorter half-life is presumably due to the presence of a disulfide bridge that is required for the covalent redox modification of PAK1 and is proposed to be essential for the PAK1 inhibitory effect of IPA-3 [238]. The inhibitory effect of IPA-3 on PAK1 activity was abolished upon dithiothreitol (DTT)-mediated reduction of IPA-3 in vitro [230]. Thus, the metabolically labile nature of IPA-3 became a major stumbling block in its use for therapeutic purposes, hence causing interest in developing IPA-3 as cancer therapeutic to cease.

Novartis developed a dibenzodiazepine-based small molecule inhibitor of PAK1, NVS-PAK1-1, that showed very high specificity for PAKs over other kinases and a ∼50x selectivity for PAK1 over PAK2 [231]. Treatment with NVS-PAK1-1 showed a trend without obvious systemic toxicity in inhibiting NF2 [239]. The efficacy of NVS-PAK1-1 on solid cancers in preclinical models is yet to be conducted. A recent study by Genentech suggested that the toxicity of group-I PAK inhibitors may be associated with the inhibition of PAK2 [240], which is an advantage for NVS-PAK1-1 over all other known PAK inhibitors. Unfortunately, PAK2 is expressed and activated in most cancers, which creates questions about the suitability of NVS-PAK-1-1 as a cancer therapeutic. A recent publication reported the development of ‘proteolysis targeting chimeric’ (PROTAC)-based NVS-PAK1-1 with superior efficacy to inhibit PAK1 activity. This PROTAC model has not been tested in vivo [241]. It has been reported that NVS-PAK1-1 has a short half-life in rat liver microsomes and, in vivo, is metabolized by the cytochrome P450 system [231,239]. Due to this, NVS-PAK1-1 and its PROTAC version may only be effective in vivo only in combination with an inhibitor of cytochrome P450 enzyme inhibitor, which likely limits their clinical utility (Table 2).

Recently, another novel class of small molecule inhibitors has been identified that displays high selectivity for PAK1 over PAK2, which is achieved by binding to the less conserved p21-binding domain at the N-terminus of PAK1 [237]. However, these compounds are allosteric PAK1 inhibitors, and PAK1 inhibitory effects are only achieved at much higher (micromolar) doses. Their efficacy in vivo and any toxicities are yet to be demonstrated. As a result, although these compounds are specific for PAK1, their clinical utility for cancer therapy is questionable.

## 11. IPA-3 as a Therapy for mPCa and PCa Cell-Induced Bone Remodeling

The majority of PCa deaths occur due to its metastasis to distant tissues such as bone [242], which results in a very low 5-year survival rate and very limited treatment options [5]. The cBioPortal database on the advanced PCa-associated bone metastasis cases [206,243] revealed significantly higher levels of PAK1 mRNA levels in patients with high Gleason scores compared to low Gleason score tissues, and in patients with PCa spread to lymph nodes and surrounding tissues, suggesting a linkage between PAK1 expression with PCa spread. In contrast, there was no difference in PAK6 mRNA expression between the normal prostate and PCa tissues or any correlation with the patient Gleason scores [198]. These findings suggest that PAK1 and not PAK6 is necessary to promote PCa metastasis thus suggesting the potential utility of PAK1 as a target for metastatic PCa.

A potential association between PAK1 expression and PCa disease severity revealed from the clinical data prompted us to investigate the role of PAK1 in advanced PCa and its causal link with PCa spread to distant tissues. Initial studies unveiled an integral role of PAK1 in the regulation of PCa EMT in vitro and the growth of tumor xenografts in vivo [196]. Confirming this, treatment with a free form of IPA-3 prevented the distant metastasis of murine advanced PCa (RM1) cells, thus demonstrating the inhibitory effect of targeting PAK1 activity on PCa invasion in vivo [195]. In a subsequent study in a clinically relevant model of PCa-induced bone remodeling by administering RM1 cells in C57BL/6 mouse tibiae [244], IPA-3 was shown to suppress the osteolytic activity induced by the RM1 cells [244]. Furthermore, RM1 cells administered into the bone also resulted in PCa metastasis to the lungs, which also was inhibited by IPA-3 treatment [244], demonstrating that IPA-3 is effective in suppressing PCa tumor growth, distant metastasis, and PCa-induced bone remodeling in these pre-clinical experimental models. These results provide for the first time, evidence that IPA-3 could be a potential treatment for patients with mPCa and even more importantly, for patients with metastasis to the bone, for whom there are very limited treatments available.

## 12. Nanoliposomal Packaging of IPA-3 for Improved Delivery, Drug Stability, and Efficacy for PCa Therapy

IPA-3 has been demonstrated to be effective in inhibiting proliferation, motility, and invasion in a variety of cancer cells such as melanoma [245,246], colon [245,246,247], breast [155], hepatocellular carcinoma [248], NSCLC [249], lung [250], HeLa [251], esophageal squamous cell carcinoma [252], and the prostate [55,155,253,254]. By targeting a protein whose expression is increased in PCa cells (PAK1), IPA-3 is expected to have greater specificity and selectivity compared to chemotherapy agents that target microtubules or DNA, since it targets the protein that is not detected in non-cancerous prostate tissue. Our studies have shown that IPA-3 effectively inhibits PCa growth, metastasis, and bone remodeling in mice [55,195,196], but only with daily administration, likely due to instability in the plasma, and has minimal effects on other cellular functions. Unfortunately, the high frequency (i.e., daily) IPA-3 administration necessary to observe IPA-3 efficacy makes it impractical for clinical use. Furthermore, since PAK1 is necessary for many physiological events [255,256,257,258], the serious toxic effects of targeting PAK1 with the free form of IPA-3 cannot be ruled out. This created the need for a novel method to deliver IPA-3 to PCa cells in vivo to improve drug stability and efficacy, minimize the frequency of drug administration, and reduce any potential side effects of free IPA-3.

Issues such as optimal dosing, drug stability, frequency of drug administration, off-target effects, and adverse reactions frequently arise in cancer clinical trials, which results in a majority of them failing [259]. Poor pharmacokinetics and drug delivery to the tumor tissues is among the most important limiting factors in the clinical development of novel cancer drugs [260]. An ideal therapeutic drug formulation for cancer must yield delivery of an effective dose of the drug specifically to the tumor tissue without inducing toxicity. Lipid-based nanoparticulate drug carriers, such as long-circulating sterically-stabilized liposomes (SSL), stably encapsulate drugs and facilitate their delivery [261,262,263]. They alter drug PK profiles, especially compared to free drugs, and can enhance their pharmacological activity [261]. Differences in the half-life and/or tissue and tumor distribution are suggested to be primary mechanisms for these actions. Additionally, SSLs can reduce off-target toxicity [264,265]. In addition, SSL accumulates passively in solid tumors due to the enhanced permeability and retention (EPR) effect mediated by defects in the vasculature and the lack of functional lymphatics [266,267]. Hence, tumor-specific drug delivery using lipid-based nanoparticles such as SSL can deliver chemotherapeutic cargo specifically to the tumor tissues with much higher efficiency compared to free drugs and other methods [268].

Initial attempts to encapsulate IPA-3 in SSL were successful with high efficiency, and the newly generated nanoliposomes (SSL-IPA-3) exhibited desirable zeta potential and particle size sufficient for their clinical utility [197]. SSL-IPA-3 suppressed PCa cell proliferation in vitro suggesting efficacy in vivo [262,269,270]. Biophysical characterization of SSL-IPA-3 showed excellent stability to encapsulate IPA-3 for at least seven days. More importantly, SSL-IPA-3 was effective in inhibiting the growth of human PC3 cell xenografts implanted in athymic nude mice with just 2 days/week administration, whereas a similar dose and frequency of administration of free IPA-3 were ineffective. This suggested that encapsulating IPA-3 in SSL improved stability and half-life, and slow release of IPA-3 by the liposomes in vivo resulted in increased efficacy.

## 13. Liposomal Formulations of IPA-3 and Their Therapeutic Benefits for mPCa

The successful development of SSL-IPA-3 prompted the generation of another IPA-3-loaded liposomal formulation for efficacy comparison with SSL-IPA-3. This formulation of liposomes was also sterically stabilized but was also designed to be responsive to secreted phospholipase A_2_ (SPRL-IPA-3), which the cancer cells release into the tumor microenvironment in abundance [271,272]. A direct comparison between SSL-IPA-3 and SPRL-IPA-3 indicated both can suppress the growth of PCa cell xenograft in athymic nude mice [194]. However, the efficacies of these two liposomal formulations in inhibiting the growth of human PCa tumor xenografts implanted in mice were very similar, and once again 2 days/week administration of free IPA-3 did not have any tumor-suppressive effect [194]. The study also demonstrated the efficacy of SSL-IPA-3 in suppressing PC3 cell metastasis to the lungs implying liposomal IPA-3 is a potential treatment strategy for mPCa. A comparison of SSL-IPA-3 with other potential future therapeutics for PCa is shown in Table 3.

## 14. Summary, Conclusions, and Future Directions

With the incidence of mPCa continuously rising over the years, there is a significant need of developing novel therapeutic approaches. This paper has reviewed the role of group-I PAKs in mPCa, the potential benefits of targeting PAK1 for mPCa therapy, the development and testing of ATP-competitive and allosteric inhibitors of PAK1 for cancer therapy, and compared pharmacologically targeting PAK1 with the other novel PCa therapies in development.

The initial steps recommended for treating clinically localized PCa include surgery and radiotherapy. Whereas ADT, androgen signaling pathways inhibitors (ASPI), and/or chemotherapy are suggested for mPCa, treatment decisions on mPCa patients will depend on whether it is newly diagnosed mPCa or progression to mPCa through non-metastatic castrate resistance [207]. Although castration provides disease control at the outset, almost all patients will eventually develop castration resistance [23]. Moreover, the lack of androgens causes significant deterioration in the quality of life, including symptoms such as obesity, fatigue, anxiety and depression, lack of concentration, loss of libido, and erectile dysfunction [21]. While immunotherapies have been a boon in the management of some cancers, this modality has not seen great success in PCa treatment, based on the outcome of several clinical trials [292] (Table 3), again, suggesting that there is a critical need for the development of new therapeutic agents to care for and manage early and advanced-stage PCa patients.

The literature strongly supports the idea of targeting PAK1 for mPCa treatment (Table 1). The absence of PAK1 in the normal prostate, its emergence in PCa, and further elevation in high-grade PCa to promote EMT and metastasis validate this hypothesis [55,196]. Cellular and preclinical studies in mice have demonstrated that pharmacological inhibition of PAK1 suppresses PCa tumor growth [55,195,196]. Despite this knowledge, the clinical development of PAK1 inhibitors for cancer therapy was a challenge. Most of the 1st generation drug candidates developed were ATP-competitive and interacted with the catalytic site of PAKs (Table 2), which are typically conserved amongst the isoforms and other kinases. The similarity in structural elements amongst kinases invariably leads to high potency, but low selectivity and high toxicity. This was the case also with these PAK1 inhibitors, which hampered their clinical development [293]. Hence, allosteric inhibitors, that cause conformational changes in protein structure and alter function by binding to sites distal from the catalytic or the ATP-binding domain were preferred for higher specificity. IPA-3 was the 1st identified PAK1 allosteric inhibitor that covalently binds to PAK1 and prevents its activation even in the presence of high levels of GTP-bound Rac and cdc42 [230]. Unfortunately, the instability of IPA-3 in the plasma due to the presence of a disulfide bond hindered its clinical development [230]. Although hopes were high with the development of NVS-PAK-1-1, a newly identified allosteric PAK1 inhibitor with very high specificity (Table 2), it quickly faded because of its short half-life in vivo, due to faster metabolism by the cytochrome P450 system [231,239]. Liposomal encapsulation (SSL-IPA-3) has given a new lease for IPA-3 by stabilizing it in the plasma and increasing half-life, which improved its efficacy to suppress PCa growth and prevent metastasis to the lungs in mice.

In conclusion, studies focusing on novel allosteric covalent modulators of group-I PAKs will be a fruitful field of future research, which will provide new opportunities to develop strategies to prevent and/or treat mPCa, independently of AR signaling modulation, and with minimal off-target effects.

## Figures and Tables

**Figure 1 cancers-15-02236-f001:**
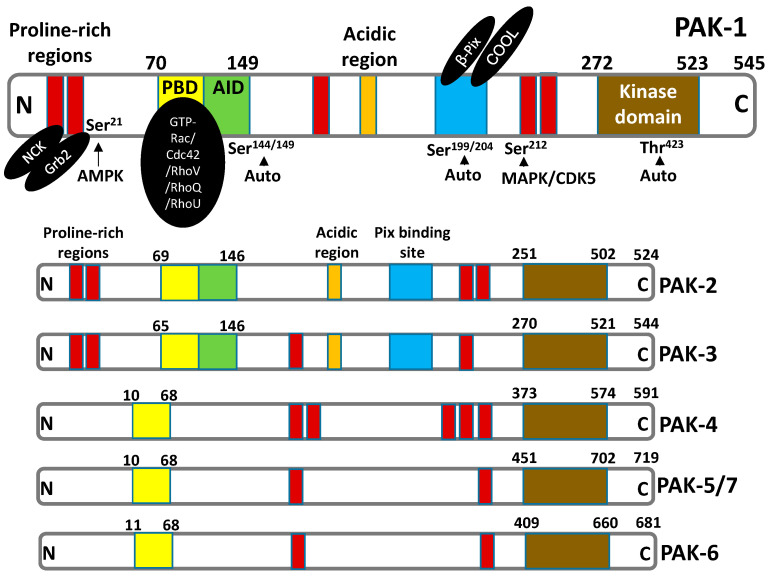
Schematic representation of the structure of human PAK isoforms. Group-I PAKs (PAK1-3) contain a conserved overlapping PBD/AID region. Interaction of small GTPases (GTP-bound Rac and cdc42) with the PBD domain releases group-I PAKs from auto-inhibition. Group-II PAKs (PAK4-6) do not contain AID, acidic and PIX-binding regions that are present in Group-I PAKs.

**Figure 2 cancers-15-02236-f002:**
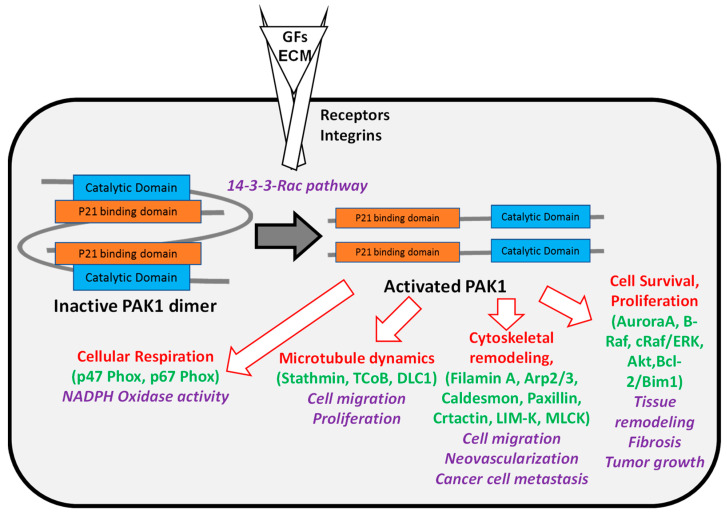
Activation of PAK1 by growth factors (GFs) and extracellular matrix (ECM) and the effect of PAK1 activation on PCa cellular function. Interaction of GTP-bound Rac/cdc42 releases PAK1 from auto-inhibition and promotes cytoskeletal dynamics, motility, invasion, proliferation, and survival in cancer cells and other cell types by phosphorylating a plethora of substrates.

**Table 2 cancers-15-02236-t002:** ATP-competitive and allosteric inhibitors of PAK1.

Compound	PAK1 Inhibition IC50	Mechanism of Action	Reference/Patent
K-252a	2.5 nM	ATP-competitive	[214]
KTD606	4.0 nM	ATP-competitive	[215]
CEP1347	2.5 nM	ATP-competitive	[215]
Staurosporine	0.75 nM	ATP-competitive	[216]
Λ-FL172	130 nM	ATP-competitive	[227]
R-1	83 nM	ATP-competitive	[232]
II-11	1.6 nM	ATP-competitive	WO 2013026914 A1. 2013
8 PF-3758309	14 nM	ATP-competitive	[217]
FRAX597	7.7 nM	ATP-competitive	[220]
FRAX486	8.3 nM	ATP-competitive	[222]
G-5555	3.7 nM	ATP-competitive	[223]
PAK inhibitor 12	65 nM	ATP-competitive	[213]
PAK inhibitor 13	5 nM	ATP-competitive	[233]
PAK inhibitor 14	5 nM	ATP-competitive	[234]
PAK inhibitor 15	288 nM	ATP-competitive	[235]
G-9791	Ki = 26 nM	ATP-competitive	[46]
PAK inhibitor 17	0.73 µM	ATP-competitive	[236]
OSU-03012	1.03 µM	ATP-competitive	[224]
AK963	----	ATP-competitive	[225]
ZMF-10	194 nM	ATP-competitive	[226]
IPA-3	2.5 µM	Allosteric	[230]
NVS-PAK1-1	4.0 nM	Allosteric	[231]
NVS-PAK1-C	2.5 nM	Allosteric	[231]
2-Mc-1,4-NHQ	0.75 nM	Allosteric	[237]

**Table 3 cancers-15-02236-t003:** Recently Approved and Future Therapies in Development for the Treatment of PCa.

Drug/Trial ID/Company	Mechanism(s) of Action	Advantage(s) and Disadvantage(s)	Reference
PD1 Integrated Anti-PSMA CART (NCT04768608)	Immunotherapy that specifically directs activated T-cells to PSMA-positive PCa cells by blocking PD1.	Whereas specificity and targeted approach are an added advantage, patients also undergo leukapheresis by receiving cyclophosphamide and fludarabine, which may have side effects. Another disadvantage is that immunotherapy is less effective for PCa.	[273]
ES414 (NCT02262910)	Immunotherapy by a humanized bispecific antibody, designed to treat mCRPC by T-cell cytotoxicity against PCa cells.	Although highly specific, its poor efficacy on PCa is a concern.	[274]
PD-L1 (NCT03179410)	Immunotherapy by targeting the ligands that bind to PD1 on T-cells.	PD-L1 antibodies are highly specific, but their efficacy on PCa is poor.	[275]
Nivolumab (NCT03040791)	Immunotherapy by specifically targeting PD1 in T-cells to prevent ligand binding,	Although highly specific in targeting T-cells, the same concerns of anti-PD-L1 and ES414 apply. Risks of developing lung, intestinal, and kidney injury are also reported.	[276]
ProstAtak (NCT01436968)	Immunotherapy + radiation therapy for patients with intermediate-high risk localized PCa. ProstAtak stimulates a cancer vaccine effect.	ProstAtak is hypothesized to improve the clinical outcome for patients with localized PCa. Its efficacy on mPCa is unclear.	[277]
^177^Lu-PSMA-617 (PLUVICTO; NCT04509557)	Selectively seeks out, attaches to PSMA on the PCa cell surface, and specifically delivers radiation to destroy PCa cells.	Whereas targeted delivery is an advantage, there are concerns about radiation exposure, bone marrow toxicity, blood cancers, kidney, liver, and hormonal gland risks, and infertility.	[278]
Rubraca (Rucaparib) and Lynparza (Olaparib)	Inhibit Poly ADP-ribose polymerase (PARP), which repairs single-strand breaks in the DNA resulting in increased double-stranded breaks and PCa cell death.	PARP inhibitors may develop bone marrow toxicity, lower the blood cell count, anemia, infection, venous thromboembolism (VTE), easy bruising/bleeding, myelodysplastic syndrome, and Acute Myeloid Leukemia.	[279]
AC0176 (Accutar Biotechnology)	An investigational orally bioavailable molecule targeting AR (phase-I study).	Although the side effects are currently unknown, targeting AR signaling will likely affect the quality of life of the patients.	[280]
Enzalutamide + Relacorilant (NCT03674814)	Enzalutamide, A specific AR inhibitor + Relacorilant, a Selective Glucocorticoid Receptor Modulator	Both drugs have serious side effects, which may have compounding effects when combined. E.g., cardiac toxicity has been reported.	[281,282]
LY2452473 (NCT02499497)	Selective AR Modulators	AR signaling will likely affect the quality of life of the patients. Other already-known side effects of androgen therapies are expected.	[283]
Darolutamide (Orion)	Oral AR antagonist	Side effects similar to other androgen therapies are expected.	[284]
ODM-208 (Orion/Merck)	A specific inhibitor of CYP11A1, an enzyme involved in androgen synthesis.	Side effects similar to other androgen therapies are expected.	[280]
EPI-7386 (Essa Pharma)	Specifically targets a specific AR variant (AR-V7), which does not respond to standard anti-androgen therapies.	Specifically designed to target mCRPC that expresses AR-V7. Although the side effects are not yet reported, adverse reactions similar to other androgen therapies are expected.	[20]
Teverelix TFA (ANTEV)	Specifically inhibits GnRH, in turn, reducing androgen synthesis	Side effects similar to other androgen therapies affecting the quality of life are expected.	[285]
IG-VMAT (NCT02934685)	Hypofractionated Image-guided Volumetric Modulated Arc Radiotherapy	Although highly efficacious, treatment is only for localized PCa, and not effective for mPCa	[286]
PF-06821497 (Pfizer)	A specific inhibitor of EZH2 (Enhancer of zeste homolog 2), a histone-lysine N-methyltransferase enzyme, resulting in transcriptional repression.	Several side effects such as anemia, myeloid suppression, hypercholesterolemia, electrolyte imbalance, shortness of breath, diarrhea, and hemorrhage have been reported.	[287]
LY01005 (NCT04563936)	Goserelin (LY01005) stops the production of testosterone and inhibits the growth of PCa cells.	Side effects are very similar to other androgen therapies.	[288]
Masitinib + Docetaxel (AB Science)	Masitinib is a tyrosine kinase inhibitor that specifically targets innate immune cells. The idea is to combine immunotherapy with a standard chemotherapy	Less efficacy of immunotherapy and the already known side-effects of taxanes are major concerns.	[289]
CAN-2409 (Candel Therapeutics)	Oncolytic viral immunotherapy kills PCa cells through rapid viral replication.	Immunotherapy in general is less effective in treating PCa. The efficacy and side effect data is pending.	Unpublished
SNX631 (Senex Biology)	A specific inhibitor of CDK8/19, which is expressed in advanced PCa.	SNX631 is currently being investigated for efficacy on PCa, and any potential side effects.	[290]
Sabizabulin (VERU-111)	Vinca alkaloid-like tubulin disruptor to suppress PCa cell proliferation.	While the efficacy and side effects are currently being investigated, targeting tubulin by taxanes and vinca-alkaloids has demonstrated serious adverse reactions.	[291]
SSL-IPA-3 or MTX-101 (MetasTx)	A highly specific allosteric inhibitor of group-I PAKs in a liposomal formulation modulates actin dynamics in cancer cells to suppress EMT.	A better safety profile is expected compared to the existing PCa therapies and is anticipated to provide high specificity and superior efficacy in preventing and treating mPCa without affecting the hormones.	[194,196]

## Data Availability

All the data are included in the manuscript.

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
