# Peer review of "Targeting P21-Activated Kinase-1 for Metastatic Prostate Cancer"

_cancers, 2023, doi:10.3390/cancers15082236_

Round 1
Reviewer 1 Report
Payaningal R. Somanath et al. reviewed the molecular mechanisms of PAKs from early stage prostate cancer cell to malignant and metastatic cancer cells. The TGFβ pathway induces the epithelial to mesenchymal transition in prostate cancer and it could activate the PAK signaling. The PAK proteins play important role in cell motility, extracellular matrix, cell survival and proliferation. Thus the PAKs maybe a potentially important target of prostate cancer therapy. The PAKs significantly increased expression level in mPCa compare to benign tissue. There are several compounds that target PAKs. The authors described the pathophysiology and current treatment guidelines of prostate cancer. Furthermore, this review discussed the development of nanotechnology-based therapeutic formulation of PAKs inhibitor and potential advantages. It is an important PAKs study in prostate cancer.
This review is well written and overall convincing. It will be of interest not only for cancer researchers but also to the cancer drug researchers. I think the manuscript should be considered for publication, as long as the authors are able to address some specific concerns (see below).
1,It is a good summary of the regulators or activators of PAKs in this manuscript, such as Rac Rac1, Rac2, and Rac3, Cdc42 as well as CHP/Wrch2/ RhoV, TC10/RhoQ, etc. I would like to suggest that using one figure to show the regulation of PAKs with these interact proteins. Perhaps, some people are interesting for the regulation of PAKs.
2,Authors mentioned that proteins interact with the PAKs, so it is better to show one figure or a table that illustrates the PAKs structure and interaction sites.
3,There are figure1 and figure2, but authors do not cite the figures properly in the manuscript.
4,This manuscript reviewed the mediators of PAKs, and the target proteins of PAKs in many different cancers, however, there is not enough specific discussion the role of PAKs in metastatic prostate cancer comparing other cancers. Because the title of the review emphasizes the PAKs in metastatic prostate cancer, it is better to study more for the importance of PAKs in metastatic prostate cancer.
5, There are some grammar problems and difficult to understand such as Increased Rac and PAK1 activity 379 associated with lymphovascular invasion and lymph node metastasis of upper urinary tract cancer 380 is also reported [185] associating with the histological grade and lymph node metastasis [186,187]. Interestingly, a study conducted in our 397 laboratory observed demonstrated that activation fo Rac1-GTPase promoted PCa cell (PC3 and 398 LNCaP)-ECM interactions, lamellipodia formation, cell migration, and trans-endothelial migration 399 [71].
6,The figure 2 is somehow too simple to show the mechanisms and proteins that interact with APKS, which were explained in the manuscript, so maybe it is better to revise the figures with more detail such as genes or pathways involved in the PAK.
7,It seems that the manuscript does not cite the table 1 in the context.
Author Response
We thank the reviewer for all the great comments regarding our review manuscript.
Comment: 1, It is a good summary of the regulators or activators of PAKs in this manuscript, such as Rac1, Rac2, Rac3, Cdc42 as well as CHP/Wrch2/ RhoV, TC10/RhoQ, etc. I would like to suggest using one figure to show the regulation of PAKs with these interact proteins. Perhaps, some people are interested in the regulation of PAKs. Comment: 2, Authors mentioned that proteins interact with the PAKs, so it is better to show one figure or a table that illustrates the PAK structure and interaction sites.
Response: We have revised Figure 1 to address the above two comments.
Comment: 3, There are figure1 and figure2, but the authors do not cite the figures properly in the manuscript.
Response: We have confirmed that we cite all the figures and tables in the manuscript.
Comment: 4, This manuscript reviewed the mediators of PAKs, and the target proteins of PAKs in many different cancers, however, there is not enough specific discussion of the role of PAKs in metastatic prostate cancer comparing other cancers. Because the title of the review emphasizes the PAKs in metastatic prostate cancer, it is better to study more for the importance of PAKs in metastatic prostate cancer.
Response: We thank the reviewer for this excellent suggestion. Unfortunately literature on mPCa is very scarce, hence we combined section 8 to the title ‘PAK1 in mPCa’. Please note that we have extensively discussed how novel therapies might help to treat mPCa in other sections.
Comment: 5, There are some grammar problems and difficult to understand such as Increased Rac and PAK1 activity associated with lymphovascular invasion and lymph node metastasis of upper urinary tract cancer is also reported [185] associating with the histological grade and lymph node metastasis [186,187]. Interestingly, a study conducted in our laboratories observed demonstrated that activation of Rac1-GTPase promoted PCa cell (PC3 and LNCaP)-ECM interactions, lamellipodia formation, cell migration, and trans-endothelial migration [71].
Response: We apologize for these errors. We have corrected them in the revised manuscript.
Comment: 6, Figure 2 is somehow too simple to show the mechanisms and proteins that interact with APKS, which were explained in the manuscript, so maybe it is better to revise the figures with more detail such as genes or pathways involved in the PAK.
Response: We have revised Figure 2.
Comment: 7, It seems that the manuscript does not cite Table 1 in the context.
Response: We have confirmed to cite all the tables in the manuscript.
Reviewer 2 Report
The paper is undoubtedly relevant for improving and developing new therapeutic approaches. However in the introduction there is no information about PSA which is the recomended diagnostic/prognostic marker for PCA, and currently to fulfill guidelines requirements some works have estimated PSA based-risk thresholds for advanced Prostate Cancer.
1) Thus I recommend to report this information on PSA( in the introduction).
2) there is a section of this work reporting about Gleason Score, anyway currently ISUP scoring is higly recommended to replace Gleason in order to improve the classification of advanced Prostate Cancer. The backgroud related to ISUP is well explained in the previous paper.
3) currently some works have developed individual nomograms to improve the prediction of" high risk prostate cancer (in cluding metastatic"in particular to apply to patients with different commorbidities, for whom the interpretation of PSA levels is challenging. I really think that this new test may improve the predictive ability of these nomograms. I invite you to consider this suggestion in the conclusions and quote this paper in which it has been recently developed a nomogram:. Ferraro S, Individual Risk Prediction of Advanced Prostate Cancer based on the combination between total Prostate-Specific Antigen (PSA) and free to total PSA ratioClin Chem Lab Med. 2023 Jan 27. doi: 10.1515/cclm-2023-0008
Author Response
We thank the reviewer for all the great comments regarding our review manuscript.
Comment: The paper is undoubtedly relevant for improving and developing new therapeutic approaches. However in the introduction there is no information about PSA which is the recommended diagnostic/prognostic marker for PCA, and currently to fulfill guidelines requirements some works have estimated PSA-based-risk thresholds for Advanced Prostate Cancer.
Response: We thank the reviewer for acknowledging the importance of our submitted manuscript. We have attempted to address all the comments by this reviewer. Please see our responses below.
Comment: 1) Thus I recommend reporting this information on PSA (in the introduction).
Response: Thanks for this comment. We have included PSA in the introduction section.
Comment: 2) there is a section of this work reporting about Gleason Score, anyway currently ISUP scoring is highly recommended to replace Gleason to improve the classification of advanced Prostate Cancer. The background related to ISUP is well explained in the previous paper.
Response: We have included ISUP scoring in section 2 of the revised manuscript.
Comment: 3) currently some works have developed individual nomograms to improve the prediction of" high-risk prostate cancer (including metastatic, in particular, to apply to patients with different comorbidities, for whom the interpretation of PSA levels is challenging. I think that this new test may improve the predictive ability of these nomograms. I invite you to consider this suggestion in the conclusions and quote this paper which has recently developed a nomogram. Ferraro S, Individual Risk Prediction of Advanced Prostate Cancer based on the combination between total Prostate-Specific Antigen (PSA) and free to total PSA ratio ‘Clin Chem Lab Med. 2023 Jan 27. doi: 10.1515/cclm-2023-0008.’
Response: We have also included this information along with the newly added ISUP guidelines in Section 2, paragraph 1.
Reviewer 3 Report
1. In section 7, I would like to see a summary table that would summarize the data on PAK1 expression in different types of cancer.
2. Is the relationship between PAK1 expression and Glisson index confirmed in patients or only in cell lines?
3. Does this article need information from Table 2? However, this article is about Targeting P21-activated Kinase-1, not prostate cancer treatment in general.
Author Response
We thank the reviewer for all the great comments regarding our review manuscript.
Comment: 1. In section 7, I would like to see a summary table that would summarize the data on PAK1 expression in different types of cancer.
Response: The suggested table on PAK1 expression in various cancers has been added to the manuscript.
Comment: 2. Is the relationship between PAK1 expression and Glisson index confirmed in patients or only in cell lines?
Response: Not just in PCa cells, a correlation between PCa Gleason grade and PAK1 expression is also observed in PCa tissues based on the data analysis using the cBioportal study. This is discussed in the 1st paragraph of the new section 9 in the revised manuscript.
Comment: 3. Does this article need information from Table 2? However, this article is about Targeting P21-activated Kinase-1, not prostate cancer treatment in general.
Response: The purpose of Table 2 (Table 3 in the revised manuscript) is to discuss the mechanisms of action, advantages, and disadvantages of the recently developed PCa treatments and novel drugs in the pipeline for PCa therapy and compare them with PAK1 allosteric inhibitors, and if targeting PAK1 can offer potential solutions to address these concerns and come up with a highly selective and effective treatment for mPCa with minimal side-effects. The table has been revised to reflect this better.
Reviewer 4 Report
The manuscript was prepared very well. The introduction section justifies the purpose of the study. I congratulate the authors for the preparation of the manuscript
I would like to congratulate the authors for the structure of the manuscript and all the research carried out. It is highly publishable. However, there are some concerns, in part important, so the review articles need revision, see below.
· What is new in this manuscript for prostate cancer?
· Include a section on strengths.
· What does this article contribute to, the authors should make their own assessment and include their own discussion of the results shown in the manuscript?
· improve the figures
· In the Conclusion section, state the most important outcome of your work. Do not simply summarize the points already made in the body — instead, interpret your findings at a higher level of abstraction. Show whether, or to what extent, you have succeeded in addressing the need stated in the Introduction (or objectives).
Author Response
Reviewer 4:
We thank the reviewer for all the great comments regarding our review manuscript.
Comment: The manuscript was prepared very well. The introduction section justifies the purpose of the study. I congratulate the authors for the preparation of the manuscript. I would like to congratulate the authors for the structure of the manuscript and all the research carried out. It is highly publishable. However, there are some concerns, in part important, so the review articles need revision, see below.
Response: We thank the reviewer for the positive comments. We have attempted to address all the comments by this reviewer. Please see our responses below.
Comment: What is new in this manuscript for prostate cancer? Include a section on strengths. What does this article contribute to, the authors should make their assessment and include their discussion of the results shown in the manuscript. Improve the figures. In the Conclusion section, state the most important outcome of your work. Do not simply summarize the points already made in the body — instead, interpret your findings at a higher level of abstraction. Show whether, or to what extent, you have succeeded in addressing the need stated in the Introduction (or objectives).
Response: We have revised the tables and figures, and expanded the conclusion section to address all the above comments by the reviewer.
Reviewer 5 Report
In this review, authors described the pathophysiology and current treatment guidelines of PCa, presenting group-I PAKs as potential target to treat mPCa patients. Moreover, authors discussed the various ATP-competitive and allosteric inhibitors of PAKs focusing on development and testing of a nanotechnology-based therapeutic formulation of group-I PAK inhibitors.
The manuscript is interesting, generally well written and illustrated. To my opinion it can be accepted in the present form
Author Response
Comment: In this review, authors described the pathophysiology and current treatment guidelines of PCa, presenting group-I PAKs as a potential target to treat mPCa patients. Moreover, the authors discussed the various ATP-competitive and allosteric inhibitors of PAKs focusing on the development and testing of a nanotechnology-based therapeutic formulation of group-I PAK inhibitors. The manuscript is interesting, generally well-written and illustrated. In my opinion, it can be accepted in its present form.
Response: We thank the reviewer for all the positive comments and the acceptance.
Round 2
Reviewer 2 Report
Accept